# Comparative Study of Mono J Single-Step Versus Two-Step Balloon Nephrostomy Placement for Urinary Tract Obstruction: Efficiency, Tolerability, and Complication Rates

**DOI:** 10.3390/jcm14124231

**Published:** 2025-06-14

**Authors:** Dimitra Akrivou, Ilias Giannakodimos, Konstantinos Adamos, Aris Kaltsas, Evangelia Mitakidi, Dimitrios Karagiannis, Michael Chrisofos, Konstantinos Skriapas, Zisis Kratiras

**Affiliations:** 1Department of Urology, General Hospital of Larissa, 41334 Larissa, Greecekostas.skriapas@hotmail.com (K.S.); 2Third Department of Urology, Attikon University Hospital, School of Medicine, National and Kapodistrian University of Athens, 12462 Athens, Greeceares-kaltsas@hotmail.com (A.K.); mxchris@yahoo.com (M.C.); kratiras.urology@gmail.com (Z.K.); 3Department of Anesthesiology, General Hospital of KAT, 14561 Athens, Greece; evangeliamitakidi@gmail.com

**Keywords:** nephrostomy, urinary tract obstruction, mono J single-step, two-step balloon nephrostomy, patient tolerance

## Abstract

**Background:** Urinary tract obstruction can lead to severe renal impairment if not managed effectively. Nephrostomy placement is a critical intervention to relieve such obstructions. The aim of this study was to compare the mono J single-step nephrostomy and the two-step balloon nephrostomy techniques in terms of procedural efficiency, patient tolerability, and complication rates. **Methods:** A prospective randomized observational design was employed, including patients requiring nephrostomy placement due to oncological conditions, urinary tract lithiasis, or functional disorders. Patients were randomized into two groups based on the different method of nephrostomy tube placement: one with the mono J single-step method and the other with the two-step balloon method. The primary outcomes were to compare the average placement time, the accuracy of the tube placement between methods, and reported complications at 3, 6, 9, and 12 months, verified through immediate post-procedural imaging. Secondary outcomes evaluated post-procedure pain levels were using the visual analogue scale (VAS) between the two methods. **Results:** The mono J single-step method had a significantly shorter average placement time (Mann–Whitney, W = 367.5, *p* = 0.006), while the two-step method demonstrated better early tube stability, as evidenced by fewer dislocations at 3 months (Chi-square, χ^2^(1) = 4.828, *p* = 0.028, Cramer’s V = 0.28) and 6 months (Chi-square, χ^2^(1) = 5.198, *p* = 0.023, Cramer’s V = 0.29). No significant differences between methods were recorded at 6 and 12 months. Patient tolerability was comparable between the two methods, and no significant difference was reported. **Conclusions:** Although the mono J single-step technique is more time-efficient, the two-step balloon method offers advantages in early tube stability, providing valuable insights for optimizing clinical decision-making in nephrostomy placement.

## 1. Introduction

Nephrostomy placement constitutes a common intervention to relieve obstruction and preserve renal function by providing an alternative pathway for urine drainage [1]. The procedure involves the insertion of a tube into the renal pelvis, allowing urine to bypass the obstructed area and drain directly from the kidney [2]. The choice of technique for nephrostomy placement can significantly impact both procedural outcomes and patient comfort [3].

Two widely used methods constitute the mono J single-step nephrostomy and the two-step balloon nephrostomy [1]. The mono J single-step nephrostomy involves a direct, single-step insertion of the nephrostomy tube, which is designed to be quick and straightforward. This method is particularly advantageous in emergency situations, while it comprises an insecure procedure, leading to higher rates of dislocation or other complications. In contrast, the two-step balloon nephrostomy method includes an initial dilation of the tract with a balloon, followed by the insertion of the nephrostomy tube [4]. This approach is theoretically beneficial in providing a more secure and stable placement of the tube, which might reduce the risk of complications such as dislocation. However, the additional step involved may increase the duration of the procedure, which could affect patient comfort and procedural efficiency [5].

This prospective randomized observational study aims to evaluate and compare the mono J single-step and two-step balloon nephrostomy methods in terms of procedural efficiency, patient comfort, and complication rates. By providing robust evidence on the relative merits and drawbacks of each technique, this study seeks to inform clinical decision-making and enhance the overall management of patients requiring nephrostomy placement.

## 2. Material and Methods

### 2.1. Study Design and Randomization

This prospective randomized observational study included patients who required nephrostomy placement due to urinary tract obstruction, focusing on those with oncology-related obstructions, urinary tract lithiasis, and various functional disorders. The study population was divided into two groups: one group undergoing a mono J single-step nephrostomy and the other a two-step balloon nephrostomy.

To ensure balanced allocation of patients and minimize selection bias, a stratified randomization method was employed based on age and the presence of clinical characteristics, such as the reason for tube placement and degree of hydronephrosis, factors that may influence outcomes following nephrostomy tube placement. Randomization was operationalized using the last digits of each patient’s electronic appointment management application (EAMA) number, which were used to assign patients in a 1:1 ratio to either the single-step (n = 33) or two-step (n = 36) nephrostomy tube placement groups. This approach allowed for an objective, reproducible method of group assignment. The randomization sequence was prepared and stratification applied independently to ensure that the distribution of age and clinical characteristics remained balanced between groups. Allocation was concealed through the use of sealed, opaque envelopes opened only at the time of assignment, further preserving the integrity of the randomization process and enhancing the internal validity of the study.

### 2.2. Inclusion and Exclusion Criteria

Inclusion criteria for eligible patients were as follows: age 18 years or older that were able to provide informed consent and needed placement of a nephrostomy tube for various reasons and having a urinary tract obstruction, due to oncological conditions or urinary tract lithiasis, or functional disorders leading to obstruction. Exclusion criteria included a history of allergic reactions to materials used in nephrostomy procedures, coagulation disorders or anticoagulant therapy that could not be safely managed, pregnancy or breastfeeding.

### 2.3. Nephrostomy Placement and Changing Procedures

This study investigated two different types of nephrostomy tube placement procedures. The mono J single-step nephrostomy involved the direct insertion of a nephrostomy tube in a single-step procedure. In contrast, the two-step balloon nephrostomy involved an initial dilation of the tract with a balloon, followed by the insertion of the nephrostomy tube.

One-step technique involved a streamlined process where the operator advances the guidewire into a renal calyx through a needle under real-time imaging guidance, using both fluoroscopy and ultrasound. After confirming proper guidewire placement, a combined dilator and catheter system is used to dilate the tract and deploy the nephrostomy tube in a single, continuous motion. The catheter is then positioned within the collecting system, ensuring the internal fixation mechanism (J-loop), and secured to the skin with a suture.

Two-step technique concerned initial access to the renal calyx using a needle and guidewire under imaging guidance. Initial access was accomplished using both ultrasound and fluoroscopy, by advancing a needle into the renal calyx under the guidance of a guidewire. Once access is confirmed, the procedure advances by enlarging the access tract in a separate step, utilizing sequential dilators over the guidewire to gradually increase the tract size. Only after the tract has been adequately dilated is the nephrostomy tube introduced over the guidewire into a renal calyx. The catheter is then positioned within the collecting system, ensuring the internal fixation mechanism (J-loop) and secured to the skin with a suture.

Patients included in the study had a regular replacement of the nephrostomy tube every 3 months. Documentation of tube changes was conducted at predetermined 3-month intervals, with final endpoints at 3, 6, 9, and 12 months. The occurrence of early dislocation events and earlier tube replacement were meticulously recorded when dislocation was identified outside the scheduled 3-month change interval, prompting immediate tube replacement.

This data was essential for assessing the stability and security of the nephrostomy placements and identifying any procedural weaknesses.

### 2.4. Assessment of Effectiveness

The effectiveness of each nephrostomy technique was assessed by measuring the time taken for placement and the accuracy of the nephrostomy tube’s location within the renal pelvis. Time measurements were meticulously recorded from the procedure’s start to the successful placement of the tube, providing a clear indication of procedural efficiency. A shorter placement time is generally desirable as it can reduce patient discomfort, the risk of complications, and the overall use of medical resources.

The accuracy of the nephrostomy tube’s placement was determined through imaging studies conducted immediately after the procedure. These studies verified the correct positioning of the tube within the renal pelvis, ensuring optimal placement to facilitate effective drainage and reduce the risk of displacement or other complications. These metrics were chosen to provide an objective measure of procedural efficiency and effectiveness, offering comprehensive evaluation and data to inform clinical practice.

### 2.5. Tolerability Assessment

Patient tolerability was a critical aspect of this study, evaluated using the International VAS score for pain. The scale consists of a straight line, typically 10 cm in length, with endpoints defining the extreme limits of pain from ‘no pain’ to ‘worst pain imaginable.’ Patients mark a point on the line representing their current pain level. All patients included in our analysis and had a nephrostomy tube placement were subjected to local anesthesia with 10 mL of lidocaine induced in the area of the puncture. No further analgesic regimen was given during the procedure. If pain remained, further analgesics were offered after pain evaluation.

Pain levels were recorded immediately after the procedure to capture the initial impact of the nephrostomy placement. Subsequent pain levels were recorded during follow-up visits to assess both short-term and long-term tolerability. These follow-up assessments, conducted at regular intervals, monitored changes in pain over time, providing a comprehensive view of patient recovery and long-term comfort. The data gathered from these pain assessments were used to calculate a mean VAS score after tube placement, offering insights into patient comfort and satisfaction, highlighting potential differences in how well patients tolerated the mono J single-step versus the two-step balloon nephrostomy.

### 2.6. Data Collection Parameters

Data were collected for several key parameters to provide a comprehensive assessment of both nephrostomy techniques. Pain levels monitored using the VAS were recorded immediately after the procedure to capture initial discomfort and at regular intervals during follow-up visits to assess ongoing pain and recovery. Patient convenience with the nephrostomy set was assessed through feedback collected within the first three months, including subjective reports on managing the nephrostomy tube and any difficulties encountered. Hospitalization time, in terms of the number of days post-procedure, was documented to assess the procedures’ impact on patient recovery times. The presence of pus, indicative of infection, was monitored during follow-up visits.

### 2.7. Statistical Analysis

To assess the difference in average placement time between the two methods, the assumptions for normality and homogeneity of variances were examined using the Shapiro–Wilk test and Levene’s test, respectively. Due to violations in the normality assumption, the non-parametric Mann–Whitney test was used for thoroughness. The outcomes of tube changes were analyzed using Chi-square tests of independence at various intervals (3, 6, 9, and 12 months) to determine any significant associations between the nephrostomy methods and the occurrence of dislocations or problem-free experiences. Cramer’s V was calculated to measure the strength of these associations. For assessing patient tolerability via VAS scores, the Wilcoxon–Mann–Whitney test was employed to compare the pain levels between the two groups, ensuring an accurate comparison despite the ordinal nature of the VAS data. Statistical analysis was conducted using R Studio 2023.12.0 and R version 4.3.2, ensuring robust and accurate data evaluation.

In order to calculate the sample size for our analysis, based on the experience and retrospective data in our hospital, we assumed that patients that undergo single-step tube placement present with a 40% risk of earlier dislocation, while patients with the two-step method with a 10% risk of early dislocation. According to this estimated difference in earlier dislocation rates, combined with statistical power of 90% and a significant level of 0.05, we calculated that 32 patients were needed in each group.

#### Ethical Considerations

The study adhered to ethical guidelines, with informed consent obtained from all participants. The study protocol was approved by the institutional review board, ensuring compliance with ethical standards in clinical research. Ethical considerations included safeguarding patient confidentiality, ensuring the voluntary nature of participation, and minimizing any potential risks associated with the study procedures.

## 3. Results

### 3.1. Baseline Characteristics

In total, 69 patients met the eligibility criteria and were included in our analysis. Thirty-three patients underwent a single-step nephrostomy tube placement, while 36 patients had a two-step nephrostomy tube placement. All patients included in our study had a benign pathologic condition for the tube placement, either stone disease or ureteral stricture, and presented with a moderate or severe hydronephrosis. Nephrostomy tube, both with one-step and two-step, were placed under both ultrasound and fluoroscopy. No failure of tube placement was reported. Baseline characteristics of included patients are described in Table 1.

### 3.2. Average Placement Time

Table 2 summarizes the average placement time for both single-step and two-step nephrostomy tube placements. Concerning the mono J single-step method, median placement time of 17 min (IQR: 15–23 min) was recorded. The two-step balloon nephrostomy had a median placement time of 21 min (IGR: 18–24 min). The analysis revealed statistically significant difference between the two methods regarding their average placement time (Mann–Whitney, W = 367.5, *p* = 0.006).

### 3.3. Tube Change Outcomes

As already mentioned, included patients had a regular nephrostomy tube replacement every 3 months and tube change outcomes were evaluated at 3, 6, 9, and 12 months. Table 3 summarizes the reported frequences of earlier dislocation and no complications between single-step and two-step tube placement methods.

At the initial 3-month period, 36% (12 patients) of included single-tube patients and 11% (4 patients) of two-step-tube patients reported earlier dislocation, respectively. This difference was statistically significant (χ^2^ = 6.16, *p* = 0.01, V = 0.28, 95% CI: 0.00–0.52), suggesting a statistically significant association between earlier tube dislocation and one-step placement method at 3 months.

During the 3–6-month period, 33% (11 patients) of included one-step-tube patients and 8% (3 patients) of two-step-tube patients reported earlier dislocation, respectively. This relationship was statistically significant (χ^2^ = 6.65, *p* = 9.90 × 10^−0.3^, V = 0.29), showing also a statistically significant association between earlier dislocation and one-step method at 6 months.

Both at the 6–9 and 9–12-month period, concerning the single-step method, 5 patients (15%) reported earlier dislocation, while 28 patients (85%) remained problem-free. Additionally, for the two-step method, 3 patients (8%) reported earlier dislocation, while 33 patients (92%) reported no problem. No statistically significant association (χ^2^ = 0.78, *p* = 0.38, V = 0.00) was reported between method placements and tube complications at 9 and 12 months, respectively.

### 3.4. Tolerability Assessment

Patient tolerability was assessed using the International VAS score for pain. A mean VAS-score was created, evaluating pain assessment of patients after their regular tube replacement. No statistically significant difference in the complaints rate between the two methods was reported (Z = 1.3867, *p* = 0.1655). The rank order of VAS score for one-step and two-step nephrostomy tube placement is summarized in Table 4.

## 4. Discussion

The findings of this study provide important insights into the comparative effectiveness, tolerability, and complication rates of mono J single-step and two-step balloon nephrostomy techniques. These results have significant implications for everyday clinical practice and patient care in the context of urinary tract obstructions. Although the decision whether to place a nephrostomy with a one-step or two-step method depends on the experience of the clinician, several other factors should be taken into consideration. Our study showed that the single-step method presents with significantly decreased placement time, constituting a reasonable solution in emergent situations in which a rapid decompression of the urinary tract is needed. This difference in procedural duration is clinically relevant, particularly in emergency settings where time efficiency is critical [6,7]. The shorter placement time of the single-step method could also reduce patient discomfort and the risk of complications associated with prolonged procedures. Of note, faster procedures can enhance operating room efficiency and reduce healthcare costs [7].

On the contrary, our study demonstrated that single-step nephrostomy tube placement is related to a statistically significant increased rate of complications in the first 3 to 6 months, compared to the two-step method. As a result, the two-step balloon method might be more suitable for patients where long-term stability of the nephrostomy tube is paramount. This includes patients with a high risk of tube dislocation, those with complex anatomical variations, or individuals requiring prolonged drainage. The early stability offered by the two-step method can reduce the need for frequent tube changes and associated complications, making it advantageous for these scenarios. However, according to our analysis, both methods are associated with similar rates of pain tolerability, showing no significant difference regarding pain perception.

Although nephrostomy tube placement has been described by Goodwin since 1954, only limited studies have provided valuable insights into the outcomes and complications associated with different PCN techniques [8]. Two predominant techniques have been described in the literature and are widely used in every-day clinical practice: the “two stick” or Sheldiger technique and the “one stick” technique [9,10]. Although the two-step technique comprises the standard and most commonly used technique, the one-step method (or direct puncture technique) constitutes an alternative approach in selected cases [6]. Compared to the standard method, the novel single-step technique is considered simple, less time-consuming with comparable postoperative complication rates to the standard techniques [7,11].

More specifically, in a retrospective study by Funaki et al., efficacy of PCN procedures with either the “single-stick” or “double-stick” techniques were reported [12]. In accordance with our findings, no significant differences between the two methods were found in terms of success rates, complications, or tube function, suggesting that the one-step technique comprises a reliable alternative solution [12]. However, in another study that compared these two procedures, the Sheldiger technique presented with a better technical success rate, especially in patients with <3 cm hydronephrosis [6]. Notably, the one-step method had significantly higher overall, major and minor complications, while both procedures were well tolerated among patients [6]. In another study presented by Wah et al., the Sheldiger technique was associated with a better technical success rate, while both methods presented comparable major and minor complications [3]. Interestingly, in contrast with our findings, the two-step technique was related to an increased incidence of tube complications, such as drainage of the catheter, dislodgement, and tube blockage [3]. In our study, the one-step method presented with increased incidence of complications, early dislocation, and a higher proportion of problem-free experiences at the initial 3 and 6 months, while no significant differences between the 9 and 12 months were reported between the two methods. These findings suggest that the initial dilation step in the two-step method may contribute to a more secure and stable placement, while both techniques ultimately provide comparable long-term outcomes. Although the two-step method may be considered more secure, non-significant differences between the 9 and 12 months may be related to the increased familiarity of patients with these tubes and better management of nephrostomy tubes after the initial 6-month period, resulting in fewer events of dislocation and a need for replacement.

Additionally, a comprehensive review of radiologically guided percutaneous nephrostomy using mixed techniques (the Seldinger or one-step method) found a high primary success rate of 96.2% within 24 h, with major complications occurring in only 0.45% of cases [10]. This supports the findings of the current study regarding the safety and effectiveness of nephrostomy procedures. A study on retrograde nephrostomy access during percutaneous nephrolithotomy reported a high stone clearance rate of 79%, with a mean operative time of 76 min and a complication rate of 3% [13]. This emphasizes the importance of procedural efficiency and the potential benefits of alternative access techniques. Furthermore, an investigation into the outcomes of emergency percutaneous nephrostomy revealed a technical success rate of 98% and a major complication rate of 6%, highlighting the procedure’s reliability in urgent settings [14]. These findings align with the current study’s results, demonstrating that the clinician should select the optimal nephrostomy method based on the emergency setting or the incidence of complications.

Currently, there are no guidelines concerning the optimal technique for tube placement and the final decision for each method depends on the urologist’s preference and experience [6]. However, several studies have proposed that the one-step method should be used in mild to severe hydronephrosis, while the two-step method should be used in all degrees of hydronephrosis [4,6]. Interestingly, the absence of hydronephrosis is related to puncture failure, especially in the one-step technique [6]. In our study, degrees of hydronephrosis were similar among patients subjected to either the one-step or two-step technique, while there were no patients without evidence of hydronephrosis.

In order to increase the familiarity and proficiency of urologists with nephrostomy tube placement, several training models have been developed for both residents and urologists. More specifically, Jeong et al. developed a porcine-based training model for ultrasound-guided nephrostomy tube placement. The satisfaction of trainees was high for both residents and consultants, and time for nephrostomy placement was significantly diminished after subsequent tries [15].

### Study Limitations

This study is subjected to several limitations that should be acknowledged. The relatively small sample size, with a total of 69 included patients, may limit the generalizability of the findings. Additionally, the study’s observational design may be subject to biases that could influence the outcomes. More specifically, patient heterogeneity in terms of underlying conditions, anatomy, and comorbidities can affect study outcomes. In addition, our study concerns a single-center experience, and combined with a relatively small sample size, could also limit the consistency and generalizability of our results. Randomized controlled trials would be beneficial to further validate these findings. Furthermore, in our study, patients were subjected to a 12-month follow-up after tube placement, and late complications may not be recorded, comprising another limitation of our study. However, shorter follow-up periods may not capture long-term complications and efficacy, while longer postoperative periods are essential to assess late complications, such as recurrent obstructions or delayed infections. Finally, our study failed to incorporate a cost-effectiveness analysis between the two procedures due to inadequate record keeping.

## 5. Conclusions

In conclusion, this study provides valuable evidence on the effectiveness, tolerability, and safety of the mono J single-step versus two-step balloon nephrostomy techniques. The findings suggest that, while the single-step method offers procedural efficiency, the two-step method provides early stability benefits, especially for the initial 6 months. Both methods are comparable in terms of patient tolerability, making them viable options for nephrostomy placement. Clinicians should consider these factors when making decisions to optimize patient outcomes and procedural success. Further well-designed, randomized studies with larger sample sizes and longer follow-up periods are needed to confirm these results and provide more robust evidence, while also providing a cost-analysis between the two procedures.

## Figures and Tables

**Table 1 jcm-14-04231-t001:** Baseline clinical characteristics of included patients.

Clinical Characteristics	One-Step Nephrostomy (n = 33)	Two-Step Nephrostomy (n = 36)	Statistical Significance (*p*-Value)
**Age (years, mean ± SD)**	61.8 ± 12.5	63.5 ± 11.2	0.68
**Gender (Male/Female, n (%))**	Male: 18 (54.5%)/Female: 15 (45.5%)	Male: 20 (55.6%) Female: 16 (44.4%)	0.92
**Body Mass Index (BMI)** **(kg/m^2^, mean ± SD)**	28.1 ± 4.8	27.6 ± 4.0	0.70
**Reason for tube placement**	Stone disease: 29 Benign stricture: 4	Stone disease: 31 Benign stricture: 5	0.75
**Degree of hydronephrosis**	Moderate: 21 Severe: 12	Moderate: 23 Severe: 13	0.80

**Table 2 jcm-14-04231-t002:** Summary of time of nephrostomy tube placement per method.

Method	Placement Time Intervals (min)			
	Median Time	Q1–Q3	Minimum Time	Maximum Time
**Single-Step**	17	15–23	12	29
**Two-Step**	21	18–24	14	28

**Table 3 jcm-14-04231-t003:** Incidence of earlier dislocation between single-step and two-step nephrostomy tube placement during 3-month follow-up periods.

Follow-Up Period (Patients, %)	Single-Step	Two-Step	Chi-Square	*p*-Value	V-Cramer	95% CI
**0–3 months**	12 (36%)	4 (11%)	6.16	0.01	0.28	0.00–0.52
**3–6 months**	11 (33%)	3 (8%)	6.65	9.90 × 10^−0.3^	0.29	0.00–0.54
**6–9 months**	5 (15%)	3 (8%)	0.78	0.38	0.00	0.00–0.32
**9–12 months**	5 (15%)	3 (8%)	0.78	0.38	0.00	0.00–0.32

**Table 4 jcm-14-04231-t004:** Mean VAS rank score after nephrostomy tube placement for one-step and two-step nephrostomy tube placement.

VAS Score	Single-Step	Two-Step
2	5	7
3	1	5
4	5	6
5	8	7
6	6	4
7	2	4
8	4	3
9	2	0

## Data Availability

The raw data supporting the conclusions of this article will be made available by the authors on request.

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
