# Peer review of "Comparative Study of Mono J Single-Step Versus Two-Step Balloon Nephrostomy Placement for Urinary Tract Obstruction: Efficiency, Tolerability, and Complication Rates"

_jcm, 2025, doi:10.3390/jcm14124231_

Round 1
Reviewer 1 Report
Comments and Suggestions for Authors
This prospective study compares the mono J single-step and two-step balloon nephrostomy techniques. The topic is clinically important, and the results are well presented. However, several major issues need to be addressed to strengthen the manuscript.
Major Points for Revision
Sample Size and Power:
The sample size (15 per group) is small, limiting statistical power and generalisability. Please provide a justification or power calculation, and discuss this limitation more clearly.
Randomisation and Methodology:
The randomisation method is not fully described. Please clarify the process and whether any stratification was used. Also, provide more details on perioperative pain management, as this could affect VAS results.
Data Interpretation:
Early tube dislocation rates differ significantly between groups, but long-term differences are not significant. Consider using survival analysis (e.g., Kaplan-Meier) for tube stability and discuss possible reasons for these findings.
Cost and Clinical Context:
The discussion would benefit from a brief cost analysis and comments on how the findings apply to different clinical scenarios (e.g., emergencies vs. elective cases).
Limitations:
Expand on the limitations, including single-centre design, small sample, and short follow-up for late complications.
Relevant Literature:
Please cite recent studies on nephrostomy training and technique, such as Jeong et al., "Training ultrasound-guided percutaneous nephrostomy technique with porcine model" (Investig Clin Urol. 2024), to provide additional context on procedural proficiency.
Minor Points
- Improve figure/table labelling and ensure terminology is consistent.
- Carefully proofread for language and formatting.
Author Response
REVIEWER 1
Comments and Suggestions for Authors
This prospective study compares the mono J single-step and two-step balloon nephrostomy techniques. The topic is clinically important, and the results are well presented. However, several major issues need to be addressed to strengthen the manuscript.
Major Points for Revision
Sample Size and Power:
The sample size (15 per group) is small, limiting statistical power and generalisability. Please provide a justification or power calculation, and discuss this limitation more clearly.
Thank you for your valuable comment. It is not well mentioned in our article, but the sample size of our study is 69 patients, 33 in single-step group and 36 in two-step group. In order to provide more clarity according to your comment, the following sentence was added in the Results section: “Totally, 69 patients met the eligibility criteria and were included in our analysis. Thirty-three patients underwent a single-step nephrostomy tube placement, while 36 patients had a two-step nephrostomy tube placement.” Also, the following text was added in the Statistical analysis section: “In order to calculate sample size for our analysis, based on the experience and retrospective data in our hospital, we assumed that patients that undergo single-step tube placement present with 40% risk of early dislocation, while patients with two-step method with 10% risk of early dislocation. According to this estimated difference in earlier dislocation rates, combined with statistical power of 90% and a significant level of 0.05, we calculated that 32 patients were needed in each group.”. The sample size (15 per group) was corrected in the Limitation section and was replaced by totally 69 patients.
Randomisation and Methodology:
The randomisation method is not fully described. Please clarify the process and whether any stratification was used. Also, provide more details on perioperative pain management, as this could affect VAS results.
Thank you for your constructive remark. The process of randomization method was added in the Material and Methods section. The following text was added: “To ensure balanced allocation of patients and minimize selection bias, a stratified randomization method was employed based on age and the presence of clinical characteristics, such reason for tube placement and degree of hydronephrosis, factors that may influence outcomes following nephrostomy tube placement. Randomization was operationalized using the last digits of each patient’s Electronic Appointment Management Application (EAMA) number, which were used to assign patients in a 1:1 ratio to either the single-step (n = 33) or two-step (n = 36) nephrostomy tube placement groups. This approach allowed for an objective, reproducible method of group assignment. The randomization sequence was prepared and stratification applied independently to ensure that the distribution of age and clinical characteristics remained balanced between groups. Allocation was concealed through the use of sealed, opaque envelopes opened only at the time of assignment, further preserving the integrity of the randomization process and enhancing the internal validity of the study.”
In addition, concerning your valuable comment about perioperative pain management, the following text was added: “All patients included in our analysis and had a nephrostomy tube placement, were subjected to local anesthesia with 10ml of lidocaine induced in the area of the puncture. No further analgesic regimens were given during the procedure. If pain remained, further analgesics were offered after pain evaluation.”
Data Interpretation:
Early tube dislocation rates differ significantly between groups, but long-term differences are not significant. Consider using survival analysis (e.g., Kaplan-Meier) for tube stability and discuss possible reasons for these findings.
Thank you for your valuable suggestion to apply survival analysis, such as Kaplan–Meier curves, to assess the temporal dynamics of nephrostomy tube dislocation. However, in our study design, all patients underwent scheduled tube replacements every 3 months, regardless of whether or not dislocation occurred. This protocol introduces a controlled intervention that resets the observation period at regular intervals, effectively limiting the utility of standard time-to-event survival models. Since each tube was only at risk for a maximum of 3 months before being replaced, either due to routine protocol or dislocation, Kaplan–Meier analysis would not accurately reflect a continuous risk period or cumulative hazard. The assumption of uninterrupted follow-up time per unit (in this case, the tube) is violated, as each replacement represents a new risk period with potentially distinct characteristics. Instead, we tried to analyze dislocation rates at each 3-month interval using categorical comparisons and contingency analysis, which better reflects the real-world structure of our data. However, as you commented, statistically significant differences were reported in 3- and 6-months interval, while no significant difference was reported in 9- and 12-month period. As a result, the following sentence “Although two-step method may be considered more secure, non-significant differences in 9 and 12 months may be related with increased familiarity of patients with these tubes and better managements of nephrostomy tubes after initial 6-months period, resulting to less events of dislocation and need for replacement.” was added in the discussion section to provided additional explanations for these findings.
Cost and Clinical Context:
The discussion would benefit from a brief cost analysis and comments on how the findings apply to different clinical scenarios (e.g., emergencies vs. elective cases).
Thank you for your interesting comment. Costs of the operative procedure in both emergent and elective, along with additional costs due to tube dislocation cases was not measured, since it was out of the scope of this review. However, the following sentence was added in the limitation section “Finally, our study failed to incorporate a cost-effectiveness analysis between the two procedures, due to inadequate record keeping.” and also the following sentence in the Conclusion section “while also providing a cost-analysis between the two procedures.” to provide its importance in future research.
Limitations:
Expand on the limitations, including single-centre design, small sample, and short follow-up for late complications.
Thank you for your constructive remark. Small sample size and short-follow-up for late complications was already mentioned in our limitation section. However, in accordance with your suggestions, the following sentence was added in the Limitation section: “In addition, our study concerns a single-center experience and combined with relatively small sample size could also limit the consistency and generizability of our results.” Ans also “comprising another limitation of our study.”.
Relevant Literature:
Please cite recent studies on nephrostomy training and technique, such as Jeong et al., "Training ultrasound-guided percutaneous nephrostomy technique with porcine model" (Investig Clin Urol. 2024), to provide additional context on procedural proficiency.
Thank you for your valuable addition. The study you suggested was cited as you proposed along with the following text: “In order to increase familiarity and proficiency of urologists with nephrostomy tube placement, several training models have been developed for both residents and urologists. More specifically, Jeong et al. developed a porcine-based training model for ultrasound-guided nephrostomy tube placement. The satisfaction of trainees was high for both residents and consultants and time for nephrostomy placement was signifi-cantly diminished after consequent tries (38197752).”
Minor Points
Thank you for your comments
- Improve figure/table labelling and ensure terminology is consistent.
Figure/Tables were corrected as you proposed, while Table 1 was added.
- Carefully proofread for language and formatting.
Language editing and formatting were made as you recommended.
Reviewer 2 Report
Comments and Suggestions for Authors
- Why pregnancy or breastfeeding were exclusion criteria? PCN is occasionally necessary in these cases.
- Nephrostomy Placement and changing Procedures - many important details are missing here (PCN set description, manufacturer, how the PCN catheter is fixed to the skin (interventional radiologists in the center where I work use catheters that have a filament that, when tightened, creates a J loop and prevents the catheter from falling out)
- Neither in the methodology, nor in the results, is shown the total number of patients included in the study in total, and in both groups also.
- Blood loss was measured during and immediately after the procedure to identify any immediate complications related to bleeding - please explain how you measured this?
- Results - It is obvious that part of the results disappeared somewhere. Overall, this section is very poorly presented, a lot of important data are missing (description of the sample size and clinical characteristics, reasons for placing PCN, PCN placement success rate, the degree of hydronephrosis, whether the PCN was placed under the guidance of EHO and/or X-ray, data on the length of wearing the PCN catheter, etc.)
- In the methodology, you specified much more data that you collected, than you presented in the results of your research. Why?
- Tube Change Outcomes - I don't understand, does it refer to the regular replacement of the PCN catheter after 3, 6, 9 and 12 months? Therefore, it should be described precisely, but in an understandable way, that you evaluated the outcomes in three-month intervals up to 12 months after the initial placement of the PCN. That's the way how to show - in the time interval of 0-3 months, 3-6, 6-9, 9-12 months.
- Tolerability Assessment - it is not clear what the table refers to - the period immediately after the procedure, or what? also, where did you show the tolerability assessments that are described in the methodology, and refer to regular visits after PCN placement?
- Discussion - must be expanded, in accordance with the relevant data that you have collected and references from the literature related to the utilization of the two PCN techniques
- Study Limitations - for the first time here in the text you specified that a total of 30 patients were included in your study!
- Overall, when trying to compare two techniques, you need to be meticulous and accurate. A lot of parameters need to be observed, and a conclusion can then be drawn from that. Your study group is insufficient for this type of research. Moreover, with the results presented like you did, it is not possible to draw any valid conclusion. Since you did not specify in the methodology in which time interval you included patients, I cannot determine how much time you will need to include more patients. My suggestion is to have a statistician help you estimate the sample size you need for this type of study comparing two PCN techniques. The topic is interesting, and this can be a very nice article. Do your best to improve it.
Author Response
REVIEWER 2
- Why pregnancy or breastfeeding were exclusion criteria? PCN is occasionally necessary in these cases.
Thank you for your interesting question. Although pregnancy and breastfeeding comprise clinical scenarios that may need a nephrostomy tube placement in cases of hydronephrosis, these conditions may be related with increased incidence of tube translocation and different perception of pain. As a result, we decided to exclude this group of patients in terms of consistency and generalizability of the results.
- Nephrostomy Placement and changing Procedures - many important details are missing here (PCN set description, manufacturer, how the PCN catheter is fixed to the skin (interventional radiologists in the center where I work use catheters that have a filament that, when tightened, creates a J loop and prevents the catheter from falling out)
Thank you for your interesting comment. Details of both techniques for nephrostomy tube placement were added in the Nephrostomy Placement and changing Procedures section as you proposed and the following text was added: “One-step technique involved a streamlined process where the operator advances the guidewire into a renal calyx through a needle under real-time imaging guidance, using both fluoroscopy and ultrasound. After confirming proper guidewire placement, a combined dilator and catheter system is used to dilate the tract and deploy the nephrostomy tube in a single, continuous motion. The catheter is then positioned within the collecting system, ensuring the internal fixation mechanism (J-loop) and secured to the skin with a suture.
Two-step technique concerned initial access to the renal calyx using a needle and guidewire under imaging guidance. Initial access was accomplished using both ultra-sound and fluoroscopy, by advancing a needle into the renal calyx under the guidance of a guidewire. Once access is confirmed, the procedure proceeds by enlarging the access tract in a separate step, utilizing sequential dilators over the guidewire to gradually increase the tract size. Only after the tract has been adequately dilated is the nephrostomy tube introduced over the guidewire into a renal calyx. The catheter is then positioned within the collecting system, ensuring the internal fixation mechanism (J-loop) and secured to the skin with a suture.”
- Neither in the methodology, nor in the results, is shown the total number of patients included in the study in total, and in both groups also.
Thank you for your constructive comment. The number of included patients and also, the number in both groups are now present in the Results section.
- Blood loss was measured during and immediately after the procedure to identify any immediate complications related to bleeding - please explain how you measured this?
Thank you for your interesting comment. The initial protocol of our study tried to evaluate blood loss during and immediately after the procedure of tube placement. However, due to to difficulties in finding objective metrics to measure intra-operative and post-operative bleeding, this collection of data was ceased and that is the reason it is not included in the Results section. Consequently, the following phrase “Blood loss was measured during and immediately after the procedure to identify any immediate complications related to bleeding.” was erased from the Methodology section.
- Results - It is obvious that part of the results disappeared somewhere. Overall, this section is very poorly presented, a lot of important data are missing (description of the sample size and clinical characteristics, reasons for placing PCN, PCN placement success rate, the degree of hydronephrosis, whether the PCN was placed under the guidance of EHO and/or X-ray, data on the length of wearing the PCN catheter, etc.)
Thank you for your constructive remark. Since the aim of our study was to compare the efficacy and safety of the two methods, along with non- significant differences among different groups, we thought that it would not be necessary to present these data. However, according to your suggestions, Section 3.1 with Baseline characteristics was added in the text. Also, Table 1 along with the text: “Totally, 69 patients met the eligibility criteria and were included in our analysis. Thirty-three patients underwent a single-step nephrostomy tube placement, while 36 patients had a two-step nephrostomy tube placement. All patients included in our study had a benign pathologic condition for the tube placement, either stone disease or ureteral stricture and presented with a moderate or severe hydronephrosis. Nephrostomy tube, both with one-step and two-step, were placed under both ultrasound and fluoroscopy. No failure of tube placement was reported. Baseline characteristics of included patients are described in Table 1.”
- In the methodology, you specified much more data that you collected, than you presented in the results of your research. Why?
Thank you for this constructive comment. In our initial protocol, described in the Methodology section, we tried to collect a variety of characteristics, such as blood loss or colic pain. However, during the study, it was really hard to collect these data or we had feed back from a small number of patients. As a result, although these data were described in the Methodology section, we managed to have data and follow-up for patients mainly for the factors presented in the Results section. According to your comments, the phrase concerning incidence of colic pain and blood loss was erased, since these data are not presented in the Results section.
- Tube Change Outcomes - I don't understand, does it refer to the regular replacement of the PCN catheter after 3, 6, 9 and 12 months? Therefore, it should be described precisely, but in an understandable way, that you evaluated the outcomes in three-month intervals up to 12 months after the initial placement of the PCN. That's the way how to show - in the time interval of 0-3 months, 3-6, 6-9, 9-12 months.
Thank you for your valuable remark. Patients changed the nephrostomy tubes regularly every 3 months. However, some patients had an earlier dislocation or a need for an earlier replacement of the nephrostomy tube, during this 3-month period. As you mentioned, we tried to evaluate the nephrostomy tube replacement at each 3-month interval. Changes were made as you proposed in the Methodology section to better express these 3-month intervals of follow up with final end-points at 3, 6, 9 and 12 months. The following text was added in the Material and Methods section: “Patients included in the study had a regular replacement of nephrostomy tube every 3 months. Documentation of tube changes was conducted at predetermined 3-month intervals, with final end-points at 3, 6, 9, and 12 months. The occurrence of early dislocation events and earlier tube replacement were meticulously recorded when dislocation was identified outside the scheduled 3-month change interval, prompting immediate tube replacement.”.
- Tolerability Assessment - it is not clear what the table refers to - the period immediately after the procedure, or what? also, where did you show the tolerability assessments that are described in the methodology, and refer to regular visits after PCN placement?
Thank you for your valuable comment. The tolerability assessment refers to the period immediately after the procedure. In addition, in order to evaluate the VAS score among patients visits, we created a mean value of VAS score after each change of nephrostomy tube. Based on your comments the following sentence was added in the Methodology section: “were used to calculate a mean VAS score after tube placement, offering…” and in the Results section: “A mean VAS-score was created, evaluating pain assessment of patients after their regular tube replacement”
- Discussion - must be expanded, in accordance with the relevant data that you have collected and references from the literature related to the utilization of the two PCN techniques
Thank you for your valuable comment the Discussion section was expanded as you proposed.
- Study Limitations - for the first time here in the text you specified that a total of 30 patients were included in your study!
The number of patients was inaccurate. The number of patients in each group is described in the Results section.
- Overall, when trying to compare two techniques, you need to be meticulous and accurate. A lot of parameters need to be observed, and a conclusion can then be drawn from that. Your study group is insufficient for this type of research. Moreover, with the results presented like you did, it is not possible to draw any valid conclusion. Since you did not specify in the methodology in which time interval you included patients, I cannot determine how much time you will need to include more patients. My suggestion is to have a statistician help you estimate the sample size you need for this type of study comparing two PCN techniques. The topic is interesting, and this can be a very nice article. Do your best to improve it.
Thank you for your valuable comment. We tried to organize a prospective randomized study, comparing single step to two step nephrostomy tube placement. Although our study is subjected to several limitation, only few studies have been published in the literature concerning the best practice for nephrostomy tube placement. The number of included patients is mostly similar to the number of other studies. Our study provides valuable evidence on the effectiveness, tolerability, and safety of mono J single-step versus two-step balloon nephrostomy techniques. The findings suggest that while the single-step method offers procedural efficiency, the two-step method provides early stability benefits, especially for the initial 6 months. As a result, we hope that with the incorporation of your suggestions in our article, our paper will be improved.
Round 2
Reviewer 1 Report
Comments and Suggestions for Authors
The revisions have satisfactorily addressed all major concerns. The paper now offers a balanced perspective on technique selection, aligned with the journal’s mission to publish impactful, evidence-based research.
Reviewer 2 Report
Comments and Suggestions for Authors
Excellent